# Patterns and Predictors of HIV Comorbidity among Adolescents and Young Adults in South Africa

**DOI:** 10.3390/ijerph21040457

**Published:** 2024-04-09

**Authors:** Brian van Wyk, Rifqah Abeeda Roomaney

**Affiliations:** 1School of Public Health, University of the Western Cape, Bellville 7535, South Africa; rifqah.roomaney@mrc.ac.za; 2Burden of Disease Research Unit, South African Medical Research Council, Cape Town 7501, South Africa

**Keywords:** HIV, adolescents, young people, co-morbidity, multimorbidity

## Abstract

Adolescents and young adults (AYA) are identified as a high-risk group for HIV acquisition. However, health services are generally not sensitive to the needs of this priority population. In addition, multimorbidity (having more than one disease in an individual) is not well studied among AYA, as it is typically associated with older individuals. This paper reports on commonly co-occurring disease conditions and disease patterns in AYA, aged 15–24 years, who took part in the 2016 South African Demographic and Health Survey. Chi-squared tests and logistic regression were used to examine the weighted prevalence of disease among those with/without HIV, and the risk factors associated with HIV. Latent class analysis (LCA) was conducted to identify common co-occurring diseases. Of the 1787 individuals included in our analysis, the weighted prevalence of HIV was 8.7%. Hypertension (30.5%), anaemia (35.8%) and diabetes (2.0%) were more prevalent among those with HIV. HIV and anaemia, hypertension and anaemia, and HIV and hypertension comprise the largest disease burden of co-occurring diseases. Co-morbidity was high among those with HIV, emphasizing the need for integrated care of HIV and non-communicable diseases.

## 1. Introduction

The Joint United Nations Programme on HIV/AIDS (UNAIDS) reported that two out of every seven new HIV infections globally occurred among adolescents and young adults (AYA) (15–24 years) in 2019 [1]. While the prevalence of HIV appears to be on a general decline, a large number of this population group are living with HIV (an estimated 3.4 million young people were living with HIV in 2019) [1]. Older adolescents (15–19 years) are considered at high risk for HIV acquisition and report poorer HIV treatment outcomes compared to other age groups [2]. However, AYA are rarely prioritized in national HIV policies and programme guidelines [3].

South Africa has a generalized HIV epidemic where the prevalence of HIV among AYA was estimated to be 4.8% in 2017 [4]. There are large disparities by sex, as girls and young women had an HIV prevalence that is nearly double that of males in the same age group (10.9% versus 4.8%, respectively) [4]. The prevalence rate of viral suppression among South African AYA is well below the revised UNAIDS target of 95% for those on antiretroviral therapy (ART) [5]. It is further reported that the burden of non-communicable diseases (NCDs) is rising in South Africa [6]. This, together with the well-established HIV epidemic, means that an increasing number of South Africans have to manage more than one chronic condition, or stated differently, are living with multimorbidity [7]. It is further argued that aging with HIV could lead to increases in HIV comorbidities and multimorbidity in younger adults, including adolescents, as long-term consequences of being on ART [8].

Multimorbidity occurs quite differently in children and adolescents when compared to adults; yet very few studies have been conducted to date to investigate the patterns and prevalence of multimorbidity in younger populations [9]. Due to the hyper HIV-epidemic in South Africa, the current paper reports on the prevalence and patterns of co-morbidity in AYA who have a known HIV status as collected in a nationally representative sample, the South African Demographic and Health Survey of 2016 (SADHS 2016).

Demographic and health surveys (DHS) have been conducted in more than 80 countries and yielded nationally representative and comparable household data that are important sources of information about health indicators and conditions in the general population (as opposed to hospital-based cohorts) [10]. The SADHS collects information on fertility levels, marriage, sexual activity, contraceptive use, nutrition, child mortality, aspects of child health, exposure to the risk of HIV infection, behaviour and health indicators as well as anthropometry, anaemia, hypertension, HbA1c levels and HIV. To date, very little research has been performed in South Africa, as in many other African countries, on multimorbidity and comorbidity in the general population.

For many countries, including South Africa, the DHS is an important evidence base for policy making, monitoring of key health conditions and national priority setting. Globally, AYA (15–24 years) are regarded as a priority population in the fight against the HIV pandemic, in addition to being the critical emerging economic workforce. It is thus important to provide evidence on the disease burden in this age group, with particular emphasis on the those living with HIV, who might be more vulnerable.

## 2. Materials and Methods

### 2.1. Survey Description

We conducted a secondary data analysis of the SADHS 2016 [11]. The survey used a stratified two-stage design. First, 750 primary sampling units (PSUs) were selected for inclusion. Second, within each PSU twenty dwelling units were randomly selected. Sub-sampling was applied and half of the households were selected for a South African-specific module on adult health that included the collection of biomarkers [11]. The SADHS is a household survey, where fieldworkers went to selected homes to administer a questionnaire and take measurements in the respondent’s home, if consent was granted. The full details of the aims and method can be found in the SADHS 2016 report [11]. For this study, we restricted the dataset to people aged 15 to 24 years of age that completed the Adult Health module.

### 2.2. Included Disease Conditions

We focused on several disease conditions based on two main criteria—we included disease conditions that were frequently included in an article which gave an overview of systematic reviews of multimorbidity [12], and disease conditions that were relevant to the South African burden of disease [13]. Additionally, we included disease conditions that could be assumed to be “current”. Two clinicians were consulted for disease conditions that were unclear. We excluded disabilities and/or injuries.

We included both self-reported diseases and conditions that were measured (e.g., biomarkers). For the self-reported disease conditions, the question was asked: “Has a doctor, nurse or health worker told you that you have or have had any of the following conditions”. We included the disease conditions: diabetes, emphysema or bronchitis, heart disease, high blood cholesterol, stroke and TB in the last 12 months. The TB in the last 12 months variable was generated by combining two other variables: whether a participant had ever had the disease and whether they had the disease in the last 12 months or more than 12 months ago.

In terms of biomarkers/measured disease, the following was of interest to the analysis: HIV status (dry blood spot), blood pressure measurements, anaemia (Hb), anthropometry (height and weight), diabetes (HbA1c). For HIV, dry blood spots were sent to a laboratory for testing [11] and the results of the first ELISA was included in this analysis.

Consenting participants provided blood samples that were drawn from a finger prick. An onsite analyser was used to determine whether the participant had anaemia (based on haemoglobin levels): mild (10.0–11.9 g/dL for adult non-pregnant women), moderate (7.0–9.9 g/dL or severe (>7.0 g/dL) [14]. We considered participants to be diagnosed with anaemia, irrespective if mild, moderate or severe.

For hypertension, the study took three blood pressure measurements using digital blood pressure monitors [11]. We excluded the first measurement and then took the average of the remaining replicated measurements. The values were categorised as: hypertension absent (systolic < 120 mmHg or diastolic < 80 mmHg), pre-hypertension (systolic: 120–139 mmHg or diastolic: 80–89 mmHg), Stage 1 hypertension: (systolic: 140–159 mmHg or diastolic: 90–99 mmHg), or Stage 2 hypertension (systolic ≥ 160 mmHg or diastolic ≥ 100 mmHg) [15]. Participants were coded as hypertension absent (normal or pre-hypertension) or present (Stage 1 or Stage 2 hypertension). Those on medication to manage hypertension were included as hypertensive.

For diabetes, dry blood spots were analysed [11] and participants were deemed diabetic if their HbA1c was ≥6.5 mmol [16,17] or if they were on medication to manage diabetes. For participants without biomarker data, we combined self-reported diabetes information. We followed the methods used in the Second South African Comparative Risk Assessment [18] for data cleaning for diabetes and hypertension.

### 2.3. Analysis

Statistical analysis was performed using Stata 15.0 (Stata Corporation, College Station, TX, USA) software, using the Stata survey set (“svy”) to account for complex survey design, and calibrating the sampling weights according to the Statistics South Africa mid-year population estimates [19].

Individuals who had more than one disease condition were assigned to the multimorbid group, following the count method used in comparative studies [20,21]. We included nine diseases of interest, namely: anaemia, bronchitis/COPD, diabetes, heart disease, high blood pressure, high cholesterol, HIV, stroke and TB in the last 12 months. The reason for including these diseases has been described elsewhere [22].

Disease variables were coded as binary (disease absent “0” or disease present “1”). Individuals were categorised as “no multimorbidity” (no disease or only one disease present) or “multimorbidity present” (two diseases or more present). Bivariate associations between locality, province, highest education level and wealth index by HIV status were assessed using Chi-square tests. The prevalence of having single disease conditions by HIV status was also assessed with Chi-square tests.

A latent class analysis (LCA) was conducted to determine disease patterns in the population using the LCA Stata Plugin [23,24]. We used the method described by Weller et al. (2020) [25]. We estimated a one-class model and in subsequent models, added additional classes. We compared the relative fit of each model and selected the model with the best relative fit using fit statistics (i.e., the model with the lowest Bayesian information criterion (BIC) [26], Akaike information criterion (AIC) [27] and adjusted BIC (aBIC)) [28]. Once the model with best fit was selected, all participants were assigned to a class based on their posterior class membership probabilities.

A multivariate logistic regression was used to determine the factors associated with having HIV. The following covariates were investigated as predictor variables: age category, sex, locality, highest education level, wealth index, employment status, BMI category, current smoker status and current alcohol drinker status.

This study was a secondary data analysis of an anonymised dataset which was obtained from the DHS programme. Ethics clearance to conduct this study was granted by the Biomedical Research Ethics Committee of the University of the Western Cape (BM20/5/8) [22].

## 3. Results

### 3.1. Sample Description

There were 10,336 males and females over the age of 15 years. After excluding those aged above 24 years of age, there were 2723 cases remaining. Participants had to give additional consent for HIV testing (and consent from a parent/caregiver if younger than 18 years). An additional 936 cases were excluded due to having unknown HIV status—meaning that they were not tested or that their HIV test results were inconclusive. Of the remaining 1787 adolescents, 8.6% were HIV positive (Table 1).

Table 1 describes the sample population by HIV status. The median age of participants was 19 years (interquartile range: 17–22 years), however those that were HIV positive tended to be slightly older (21 years, IQR: 18–23). There were marginally more females (52.3%, n = 935) than males in the population; however, among those that were HIV positive, 81.1% (n = 124) were females. More participants lived in rural areas (54.8%, n = 980). Almost 30% of people with HIV were from KwaZulu-Natal province. Most (83.4%, n = 1491) had completed secondary education. There were no significant differences in employment status by HIV status. This could be expected as very few were employed, which makes sense for this young age group. For the HIV positive group, 35.3% belonged to the poorest wealth quintile. Appendix A describe the sample population by sex and age-group, respectively.

### 3.2. Prevalence of Single Diseases within Population

Table 2 shows the weighted prevalence of each included disease condition by HIV status. HIV was prevalent in this population at 8.37% and was more prevalent among females compared to males (13.7% versus 3.7%, respectively) (Appendix A). Diabetes, heart disease, stroke, TB, hypertension and anaemia were higher in those who were HIV positive.

Diabetes was the only disease condition included in our study that was both physically measured and self-reported in the questionnaire. The prevalence of physically measured diabetes was higher than that of self-reported diabetes (0.2% versus 1.6%, respectively). This indicates that self-reported diabetes is most likely underreported, i.e., the participants were unaware that they may have the disease or chose not to disclose that they had the disease. When combining the responses of the measured and self-reported diabetes, the composite prevalence was 1.2%. Anaemia was more common in females, while hypertension was more common in males (Appendix A). Appendix A reports disease conditions by age group.

The number of diseases in individuals was also assessed by HIV status (Table 3). Having two or more diseases was far more common in those that were HIV positive, with HIV co-morbidity being more common than only having HIV. Appendix A report the number of disease conditions by age and sex, respectively.

A logistic regression was conducted to determine the lifestyle and risk factors associated with having HIV in this age group (Table 4). The regression showed that belonging to the older age group (age 20–24 versus 15–19) more than doubled the odds of being HIV positive. Being female was associated with being HIV positive (OR: 3.84, 95% CI: 2.38–6.17). Living in an urban locality was also associated with being HIV positive. Belonging to the wealthier quintiles and being employed reduced the odds of being HIV positive.

A latent class analysis was conducted to determine disease patterns within young people. Only those with one of the diseases reported in Table 2 were included (n = 159). A five-class model was found to be most appropriate as five classes had the lowest AIC and adjusted BIC values (Appendix A).

The largest disease class was hypertension and anaemia (Class 4), with one-third of the population falling into this group (Figure 1). HIV prominently featured in two disease classes, namely Class 2 (HIV and anaemia) which accounted for 33.2% of the population, and Class 3 (HIV and hypertension) which accounted for 19.0% of the population. The response probabilities of disease by each class can be found in Appendix A.

## 4. Discussion

This study aimed to determine the disease patterns present in adolescents and young adults from a national survey in South Africa. We found that about one-quarter of young people had one disease, and 5% had two or more diseases. Anaemia (23.3%), hypertension (18.6%) and HIV (8.4%) were the most prevalent single diseases in the population.

We found that having HIV was associated with belonging to the older age group (20–24 years), being female and living in an urban area. Those with tertiary education, employed and belonging to the richer and richest wealth quintiles had lower odds of being HIV positive. When looking at those that were HIV-positive, more than half of young people had one or more additional diseases. All disease conditions examined were higher in those with HIV. Although few studies have examined multimorbidity in young people with HIV, one cross-sectional study undertaken among 92 adolescents and young adults aged 15–24 years receiving treatment for HIV in primary care settings in Cape Town found that only 5% had hypertension [29]. They also found high levels of NCD risk factors such as food insecurity, low fruit and vegetable consumption and low physical activity levels.

Hypertension was much higher in people living with HIV in our study (30.5%). However, a study in Uganda found a similar prevalence (27%) among young people (aged 13–25 years) living with HIV, highlighting a high burden in elevated blood pressure among this group [30]. Other studies have also reported that hypertension and pre-hypertension are common among the population of sub-Saharan Africa [31]. However, access to treatment and monitoring of hypertension among younger adults appears to be limited in most sub-Saharan African countries.

In addition, we conducted a latent class analysis to determine common disease classes in the multimorbid population in this age group. We identified five common co-occurring disease classes: anaemia and hypertension (33.4%), HIV and anaemia (33.2%), HIV and hypertension (19.0%), hypertension and heart disease (7.8%), and diabetes and hypertension (6.6%). Similar latent classes were identified in a study that included all people over the age of 15 years also using the SADHS 2016 (e.g., (1) HIV, hypertension and anaemia (39.4%), (2) anaemia and hypertension (23.7%), (3) cardiovascular-related (19.9%), and (4) diabetes and hypertension (17.0%)) [32], indicating that these diseases could start clustering together at fairly young ages in individuals. However, longitudinal studies may be more useful in determining the age of onset for these diseases.

This study was limited in several ways. For one, we used a mix of self-reported and measured disease outcomes. The prevalence of the measured disease outcomes was far higher than those that were self-reported. This could indicate a real difference (e.g., stroke, high cholesterol and heart disease could be low in this population) or it could show that there may have been young people unaware of their disease status in the case of self-reported conditions. As the study is cross-sectional, we also cannot determine causality nor make assumptions about which disease developed first among the multimorbid population. Another limitation is the lack of information on mental health disorders, which may be prevalent among young people [33].

An additional limitation of the study is that we were not able to categorize people by the length of time they were infected with HIV and/or were on ART. Thus, it is possible that some adolescents and young adults were born or infected with HIV when young and that their co-morbidities may have differed from people who were more recently infected with HIV [34].

While self-reported information is often underreported as people may be unaware that they have a health condition, an additional bias may exist in self-reported disease outcomes among people with HIV compared to those without HIV. For example, self-reported disease outcomes could be higher among people who are diagnosed with HIV due to frequent or regular contact with healthcare providers. This may be in comparison to relatively “healthy” people of the same age who may rarely access medical treatment and are thus unaware of health conditions that they may have.

A review was conducted on NCD policies among adolescents and found that, globally, many effective interventions exist but that these often do not include contraceptive use, drug harm reduction, mental health and nutrition [33]. The review concluded that multi-sectoral efforts are needed to mitigate NCD risk factors, burden and adverse adulthood outcomes [33]. Given the high HIV burden in South Africa, and existing strategies targeting HIV prevention, we argue that NCD risk factor prevention should also be considered in existing campaigns. In addition, adolescents and young adults with HIV should be regularly screened for other diseases such as anaemia and hypertension.

## 5. Conclusions

This study makes important contributions to the field in terms of NCDs and HIV data on adolescents and young adults in South Africa. Since these data are from 2016, it can be assumed that, at present, the youngest in this dataset may already be around 22 years old and the oldest in the dataset would be around 31 years old. As these young individuals have entered adulthood, the prevalence of NCDs and HIV would be expected to increase in their cohort—with possible negative economic consequences for individuals (e.g., job loss, absenteeism), adverse outcomes in pregnancy, and generally negative effects throughout the life course.

More effort is needed in targeting adolescents and young adults in the prevention of HIV and NCDs. In addition, adolescents and young adults living with HIV, should be prioritized for individualized screening for other diseases, particularly NCDs, which could complicate their HIV treatment. Health services in South Africa and the rest of the continent should be oriented to provide comprehensive care to all people living with HIV to ensure that the burden of multimorbidity is appropriately managed.

## Figures and Tables

**Figure 1 ijerph-21-00457-f001:**
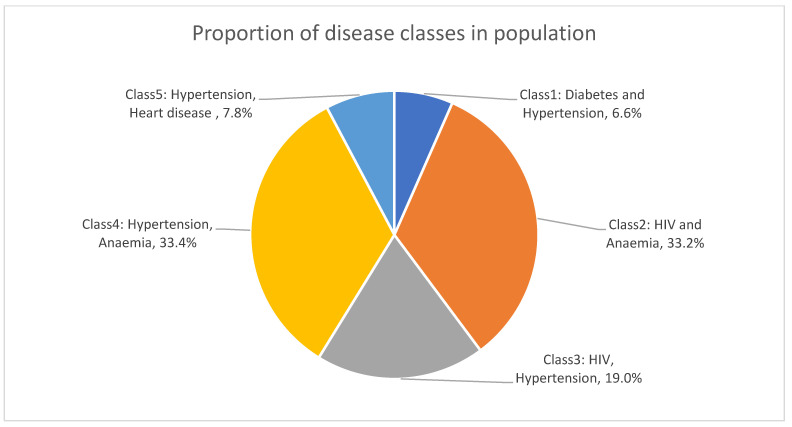
Proportion of disease classes in population.

**Table 1 ijerph-21-00457-t001:** Description of sample population by HIV status (unweighted).

	Total % (n) (n = 1787)	HIV Status % (n)		*p*-Value
		HIV Positive (n = 153)	HIV Negative (n = 1634)	
Age * (Median years and IQR)	19 (17–22)	21 (18–23)	19 (17–22)	<0.001 *^,^∞
Gender				<0.001 *
− Male	47.7 (852)	19.0 (29)	50.4 (823)	
− Female	52.3 (935)	81.1 (124)	49.6 (811)	
Locality				0.028 *
− Rural	54.8 (980)	46.4 (71)	55.6 (909)	
− Urban	45.2 (807)	53.6 (82)	44.4 (725)	
Province				<0.001 *
− Western Cape	3.5 (63)	2 (3)	3.7 (60)	
− Eastern Cape	16.0 (285)	19 (29)	15.7 (256)	
− Northern Cape	6.0 (107)	3.3 (5)	6.2 (102)	
− Free State	11.2 (201)	13.1 (20)	11.1 (181)	
− Kwa-Zulu Natal	16.8 (300)	29.4 (45)	15.6 (255)	
− North West	11.8 (211)	8.5 (13)	12.1 (198)	
− Gauteng	6.4 (114)	5.9 (9)	6.4 (105)	
− Mpumalanga	14.7 (263)	15.7 (24)	14.6 (239)	
− Limpopo	13.6 (243)	3.3 (5)	14.6 (238)	
Education level				0.225
− Primary or less	11.8 (210)	12.4 (19)	11.7 (191)	
− Secondary complete	83.4 (1 491)	85.6 (131)	83.2 (1360)	
− Tertiary	4.8 (86)	2 (3)	5.1 (83)	
Employment status				0.115
− Not employed	87.5 (1 563)	91.5 (140)	87.1 (1423)	
− Employed	12.5 (224)	8.5 (13)	12.9 (211)	
Wealth index				0.062
− Quintile 1 (Poorest)	26.5 (473)	35.3 (54)	25.6 (419)	
− Quintile 2 (Poorer)	22.6 (404)	21.6 (33)	22.7 (371)	
− Quintile 3 (Middle)	24.9 (444)	24.8 (38)	24.9 (406)	
− Quintile 4 (Richer)	18.9 (337)	13.7 (21)	19.3 (316)	
− Quintile 5 (Richest)	7.2 (129)	4.6 (7)	7.5 (122)	
BMI				0.092
− Underweight	12.1 (214)	8.7 (13)	12.4 (201)	
− Normal weight	61.9 (1095)	56.0 (84)	62.4 (1011)	
− Overweight	17.6 (311)	24.0 (36)	17.0 (275)	
− Obesity Grade 1	5.5 (97)	6.7 (10)	5.4 (87)	
− Obesity Grade 2	2.2 (38)	4.0 (6)	2.0 (32)	
− Obesity Grade 3	0.9 (15)	0.7 (1)	0.9 (14)	
Current Smoker (Yes)	12.1 (216)	5.9 (9)	12.7 (207)	0.014
Current Alcohol Consumption (Yes)	35 (625)	28.8 (44)	35.6 (581)	0.092

* *p*-value < 0.005. ∞ Kruskal–Wallis test used for numerical variable age. Chi-square test used for categorical variables.

**Table 2 ijerph-21-00457-t002:** Prevalence of single disease conditions by HIV status and method of measurement in South Africa for 2016 (weighted data).

Disease Condition		Total % (95% CI)	Prevalence by HIV Status % (95% CI)	
			HIV Positive	HIV Negative
Self-reported	Diabetes	0.22 (0.09–0.52)	0.44 (0.06–3.07)	0.20 (0.08–0.50)
	Bronchitis/COPD	0.27 (0.14–0.53)	0	0.25 (0.10–0.62)
	Heart disease	0.51 (0.29–0.88)	1.51 (0.46–4.89)	0.45 (0.18–1.10)
	High cholesterol	0.47 (0.2–1.09)	0	0.71 (0.27–1.86)
	Stroke	0.1 (0.04–0.28)	0.30 (0.04–2.16)	0.07 (0.02–0.32)
	TB (in past 12 months)	0.47 (0.26–0.84)	1.62 (0.58–4.47)	0.44 (0.18–1.04)
Measured	HIV	8.37 (6.8–10.27)	100	-
	Hypertension	18.56 (16.33–21.03)	30.54 (21.49–41.39)	17.04 (14.68–19.70)
	Anaemia	23.27 (20.98–25.73)	35.84 (27.36–45.30)	21.14 (18.65–23.87)
	Diabetes	1.61 (1.04–2.49)	2.03 (0.69–5.80)	1.12 (0.69–1.82)
Self-reported and measured	Diabetes	1.18 (0.78–1.76)	2.03 (0.69–5.80)	1.28 (0.82–2.00)

**Table 3 ijerph-21-00457-t003:** Number of diseases by HIV status.

Number of Diseases	Total	Prevalence by HIV Status % (95% CI)	
		HIV Positive	HIV Negative
No diseases	69.64 (67.11–72.05)	NA	63.79 (60.70–66.76)
One disease	25.13 (23.04–27.35)	46.71 (37.52–56.13)	32.62 (29.91–35.46)
Two or more diseases	5.23 (4.25–6.41)	53.29 (43.87–62.48)	3.59 (2.59–4.97)

**Table 4 ijerph-21-00457-t004:** Sociodemographic and lifestyle risk factors associated with HIV.

Covariates	Odds Ratio	Standard Error	z	*p* > z	(95% CI)
Age 20–24 years (Ref: 15–19 years)	2.54 *	0.49	4.78	0	1.73–3.72
Sex (Ref: male)	3.84 *	0.93	5.54	0	2.38–6.17
Locality (Ref: rural)	2.17 *	0.45	3.74	0	1.45–3.26
Education (Ref: primary)					
− Secondary	0.80	0.22	−0.81	0.418	0.46–1.38
− Tertiary	0.26 *	0.17	−2.03	0.042	0.07–0.95
Wealth Index (Ref: poorest)					
− Poorer	0.70	0.17	−1.46	0.144	0.43–1.13
− Middle	0.62	0.15	−1.92	0.055	0.38–1.01
− Richer	0.33 *	0.10	−3.55	0	0.18–0.61
− Richest	0.31 *	0.14	−2.58	0.01	0.13–0.75
Employed	0.52 *	0.17	−2.04	0.042	0.28–0.98
BMI Category					
− Normal weight	0.79	0.26	−0.74	0.462	0.42–1.49
− Overweight	0.78	0.29	−0.67	0.503	0.38–1.60
− Obesity Grade 1	0.63	0.30	−0.99	0.324	0.25–1.58
− Obesity Grade 2	1.00	0.57	−0.01	0.994	0.33–3.03
− Obesity Grade 3	0.41	0.45	−0.81	0.417	0.05–3.57
Current alcohol use	1.10	0.23	0.47	0.64	0.73–1.65
Current smoker	0.62	0.24	−1.23	0.218	0.29–1.33

* Significant at *p* < 0.05.

## Data Availability

The DHS 2016 can be requested from the DHS Programme. Analysis conducted for this paper is available on request.

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
