# Peer review of "Patterns and Predictors of HIV Comorbidity among Adolescents and Young Adults in South Africa"

_ijerph, 2024, doi:10.3390/ijerph21040457_

Round 1

Reviewer 1 Report

Comments and Suggestions for Authors

Reviewer 2 Report

Comments and Suggestions for Authors

Please explain why you exclude 936 cases due to unknown HIV statuses. While you tested the HIV anyway.

Line 128-129: Of the remaining 1,781 adolescents, 8.6% 129 were HIV positive. Did you mean 1787?

Line 156-158: The prevalence of physically measured diabetes was higher than that of self-reported diabetes (0.2% versus 1.6%, respectively). This indicates that self-reported diabetes is most likely underreported. How were the authors sure about this? The respondent could also really not know about their condition of having diabetes, not because they did not report it. 

Please explain in the method section how you measured the biomarkers/measured disease. Did you invite the respondent to the hospital? Or did you go to their houses to collect the blood samples and measurements?  

Reviewer 3 Report

Comments and Suggestions for Authors

In this South African study, the authors included 1787 people aged 15-24 that had been previously included in the national survey SADHS 2016. The aim of the study was to assess the burden of comorbidity in people aged 15-24 categorized by HIV-status. As one would expect the most common diagnoses in this young cohort, besides HIV, was anemia and hypertension. Also, as shown in other studies HIV was associated with older age, female sex and urban living.

I have a few comments/questions:

1.        Do you have any data regarding who of the people with HIV that were born with HIV? The odds of comorbidity are likely different in young people who have lived their whole life with HIV and been on ART for many years and those that were more recently sexually infected.

2.        The prevalence of hypertension was unexpectedly high. I recommend adding to the Discussions any hypothesis to why this was seen.

3.        I suggest adding to Discussion/limitations some thoughts on any selection bias regarding self-reported comorbidity. One would assume that those with HIV, that regularly see doctors/nurses are more likely to have been diagnosed with any comorbidity than HIV-negative young people that probably seldom seek health care.

Round 2

Reviewer 1 Report

Comments and Suggestions for Authors

The authors revised the comment accordingly and make sufficient improvement. However, the similarity index of 30% is a great concern, I hope the author could address this issue promptly. TQ

Round 3

Reviewer 1 Report

Comments and Suggestions for Authors

Acknowledge your explanation.